# Utility of Methicillin-Resistant *Staphylococcus aureus* Nares Screening in Hospitalized Children with Acute Infectious Disease Syndromes

**DOI:** 10.3390/antibiotics10121434

**Published:** 2021-11-23

**Authors:** Ashley Sands, Nicole Mulvey, Denise Iacono, Jane Cerise, Stefan H. F. Hagmann

**Affiliations:** 1Division of Pediatric Infectious Disease, Cohen Children’s Medical Center of New York, Northwell Health, Long Island, NY 11040, USA; shagmann@northwell.edu; 2Department of Pharmacy, Plainview Hospital, Northwell Health, Plainview, NY 11803, USA; nmulvey@northwell.edu; 3Department of Pharmacy, Cohen Children’s Medical Center of New York, Northwell Health, Long Island, NY 11040, USA; Diacono1@northwell.edu; 4Biostatistics Unit, Feinstein Institutes for Medical Research, Northwell Health, Long Island, NY 11030, USA; jcerise@northwell.edu; 5Donald and Barbara Zucker School of Medicine at Hofstra, Northwell Health, Hempstead, NY 11549, USA

**Keywords:** MRSA, screening, infections in children

## Abstract

Studies in adults support the use of a negative methicillin-resistant *Staphylococcus aureus* (MRSA) nares screening (MNS) to help limit empiric anti-MRSA antibiotic therapy. We aimed to evaluate the use of MNS for anti-MRSA antibiotic de-escalation in hospitalized children (<18 years). Records of patients admitted between 1 January 2015 and 31 December 2020 with a presumed infectious diagnosis who were started on anti-MRSA antibiotics, had a PCR-based MNS, and a clinical culture performed were retrospectively reviewed. A total of 95 children were included with a median age (range) of 2 (0–17) years. The top three diagnosis groups were skin and soft tissue infections (*n* = 38, 40%), toxin-mediated syndromes (*n* = 17, 17.9%), and osteoarticular infections (*n* = 14, 14.7%). Nasal MRSA colonization and growth of MRSA in clinical cultures was found in seven patients (7.4%) each. The specificity and the negative predictive value (NPV) of the MNS to predict a clinical MRSA infection were both 95.5%. About half (*n* = 55, 57.9%) had anti-MRSA antibiotics discontinued in-house. A quarter (*n* = 14, 25.5%) were de-escalated based on the negative MNS test alone, and another third (*n* = 21, 38.2%) after negative MNS test and negative culture results became available. A high NPV suggests that MNS may be useful for limiting unnecessary anti-MRSA therapy and thereby a useful antimicrobial stewardship tool for hospitalized children. Prospective studies are needed to further characterize the utility of MNS for specific infectious diagnoses.

## 1. Introduction

Methicillin-resistant *Staphylococcus aureus* (MRSA) is a common cause of both localized and invasive suppurative infections as well as several toxin-mediated syndromes in children. This has led to the frequent inclusion of anti-MRSA antimicrobials in empiric management plans for pediatric patients treated for presumed infections. As anti-MRSA antibiotics can lead to various toxicities, including acute kidney injury and *C. difficile* colitis, discontinuing unnecessary anti-MRSA antibiotics represents a central antibiotic stewardship priority [1]. 

While it has been recognized that carriage of *S. aureus* is an important risk factor for developing a subsequent clinical infection, the nares, in particular, were noted as the most frequent carriage site for *S. aureus* [2]. Consequently, multiple clinical studies in adult patients, especially with pneumonia, have demonstrated the usefulness of a negative MRSA nares screening (MNS) test in aiding the de-escalation of anti-MRSA antimicrobials [3,4,5,6]. Moreover, it has been demonstrated that MNS is a valuable antimicrobial stewardship tool with potential applications beyond lower respiratory tract infections, including patients with bloodstream, abdominal, and skin and soft tissue infections [7]. 

Studies in pediatric patients have thus far focused mostly on determining *S. aureus-* and MRSA-colonization rates, especially in patients receiving care in the neonatal intensive care environment as well as pre-operatively [8,9]. Studies evaluating MNS in children admitted for presumed infectious disease diagnoses are lacking. 

Our study aimed to retrospectively analyze the utility of MNS in hospitalized children with presumed infectious conditions, with the goal of comparing MNS with clinical culture results. We additionally looked at what impact, if any, MNS results had on antimicrobial choice and duration. 

## 2. Results

A total of 95 pediatric patients admitted for an infectious diseases syndrome during the study period fulfilled the inclusion criteria including MNS, submitted clinical culture and empiric start of an anti-MRSA antibiotic. The median age (range) was 2 years (0–17), 57.9% (*n* = 55) were male, and 42.1% (*n* = 40) were female. Most common diagnosis groups were skin and soft tissue infections (SSTI) (*n* = 38, 40%), toxin-mediated syndromes (*n* = 17, 17.9%), and osteoarticular infections (OAI) (*n* = 14, 14.7%) (Table 1). Most were started on clindamycin (*n* = 55, 57.9%) and vancomycin (*n* = 43, 45.3%) (Table 1). Two-thirds (*n* = 63, 66.3%) had clinical cultures from infectious sites, 44.2% (*n* = 42) had blood cultures, and 10.5% (*n* = 10) had both blood and infectious site cultures obtained. 

Seven patients (7.4%) screened positive for MRSA via MNS. 

Fifty-three (55.8%) patients had clinical cultures that showed growth. The most common organism isolated was methicillin-sensitive *Staphylococcus aureus* (MSSA) (n = 23, 24.2%), followed by *Streptococcus* spp. (n = 14, 14.7%), and MRSA (n = 7, 7.4%). Patients with toxin-mediated syndromes were most likely to have a positive clinical culture (70.6%), followed by patients with oto/mastoiditis (66.7%) (Table 2). 

Eighty-seven patients showed concordance between MNS and clinical culture, while eight patients had discordant results (Table 3). The three patients who had both positive MNS and MRSA identified from clinical cultures were one patient with sepsis, tracheitis, and *Staphylococcus* scalded skin syndrome, respectively. The four patients with negative MNS but clinical cultures positive for MRSA had diagnoses of preseptal cellulitis (*n* = 1), *Staphylococcus* scalded skin syndrome (*n* = 1), and bacteremia (*n* = 2). Of the four patients with a positive MNS and clinical cultures negative for MRSA, two were admitted with SSTI, one each with mastoiditis and a toxin-mediated illness. The sensitivity of MNS to predict an MRSA infection was 42.9%, with a specificity of 95.5%. The positive predictive value and negative predictive value (NPV) were 42.9 and 95.5%, respectively (Table 4).

A total of 55 (57.9%) had anti-MRSA antimicrobials discontinued during their hospitalization. Clinical reasoning for discontinuation were negative MNS and no growth in clinical cultures (*n* = 21, 38.2%), growth of an alternative pathogen in clinical cultures (*n* = 17, 30.2%), and negative MNS alone (*n* = 14, 25.5%). The respective median duration and interquartile range (IQR: 25th–75th percentile) of anti-MRSA antibiotic use were 2.0 (1.0–3.0), 3.0 (2.0–3.0), and 2.0 (1.0–3.0) days, respectively (*p* = 0.155 by Kruskall–Wallis rank sum test). Three patients had anti-MRSA antibiotic discontinued for reasons not related to MNS or clinical culture results and were thus not included in the Kruskal–Wallis rank sum test.

## 3. Discussion

In our study of pediatric patients hospitalized for a presumed infectious diagnosis and started on empiric broad-spectrum antimicrobial treatment including anti-MRSA antibiotics, an MNS test, when correlated with clinical cultures, was found to have a high NPV for a clinical MRSA infection. Similar to previous analyses in adults that are promoting MNS as a potentially powerful stewardship tool for de-escalation and avoidance of empirical anti-MRSA therapy [3,4,5,6,7], our findings suggest that MNS could also be a useful tool in limiting anti-MRSA antimicrobials in pediatric patients.

It needs to be noted, however, that the performance of the MNS (i.e., high NPV) in predicting clinical MRSA infection is intimately related to a low prevalence of MRSA infection in the community [10]. While it has been shown that a large proportion of children are intermittently colonized with *S. aureus*, our study, like others, has found only a minority (7.4%) of children being colonized with MRSA [11,12,13]. Moreover, of the children in this study with documented growth of *S. aureus* in clinical culture, MSSA dominated over MRSA (MSSA, 23/30 (76.7%) vs. MRSA, 7/30 (23.3%)), as has also been shown in a recent study of children with invasive *S. aureus* infections in which MRSA accounted for only one in five of the infections [14]. Together those findings are suggestive of a low background prevalence of MRSA infection at our institution and hence may support the utility of the MNS in a clinical context.

While best evidence for the utility of MNS in predicting MRSA infection exists for adult patients with pneumonia [6], more recent adult studies also demonstrated its usefulness in the context of skin and soft tissue, bloodstream, intraabdominal, and urinary infections [7]. Given our limited sample size, diagnosis-group specific analysis of MNS characteristics could not be performed, and our cohort notably included very few patients with pneumonia (*n* = 6). This was likely secondary to the circumstance that a large proportion of patients included in this retrospective analysis were admitted during the early phase of the COVID-19 pandemic when overall pediatric admissions due to respiratory tract infections had declined [15,16]. Of note, two of the seven patients with an MRSA infection had been bacteremic while the MNS were negative, raising the concern about the MNS’s performance in pediatric patients in suspected sepsis. 

In adult studies, the downstream effects of avoiding or stopping anti-MRSA antibiotics early because of a negative MNS have been manifold, including decreased length of anti-MRSA antibiotics, fewer drug-monitoring labs, decreased nephrotoxicity, and decreased length of stay [3,4,5]. In this context, we took great interest to see that the decision to discontinue the anti-MRSA antimicrobial treatment was supported by a negative MNS alone, or by a negative MNS in combination with a negative clinical culture in a quarter and about a third of those children in whom treatment was stopped during hospitalization. Of additional importance, the point estimates for the duration of anti-MRSA antibiotics were the shortest among the children who had negative MNS when compared to those children in whom the decision to discontinue anti-MRSA antibiotics was based on a positive clinical culture result with an alternative pathogen. The difference, however, did not reach statistical significance, which may have been secondary to the small sample studied. 

This study has several limitations. Besides the retrospective design and small sample size, which precluded diagnosis group-specific analysis, a major limitation was that the determination of the reason for discontinuation of anti-MRSA antibiotics was subject to the reviewers’ interpretation of the clinical records. Further, it is possible that the nasal MRSA colonization status was misclassified in some patients because MNS had been obtained after the initiation of anti-MRSA antibiotics. This effect, however, may have been minimal as patients on anti-MRSA antibiotics >48 h prior to collection of MNS were excluded and a PCR assay was used for the MNS, which is more likely to remain positive in patients already receiving antibiotic therapy [17]. 

In conclusion, our study shows that clinicians at a tertiary children’s hospital have started to use a PCR-based MNS to help manage empiric anti-MRSA antibiotics in children admitted for infectious syndromes. Similar to the evidence found in adult studies, we demonstrated that a negative MNS had a high NPV for clinical MRSA infection and consequently could serve as a helpful antibiotic stewardship tool for avoiding and limiting unnecessary anti-MRSA antibiotic treatment. Prospective studies to evaluate the utility of MNS for various infectious diagnosis groups are warranted.

## 4. Materials and Methods

This single-site retrospective study was conducted at Cohen Children’s Medical Center/Northwell Health, a tertiary children’s hospital in the metropolitan New York City region. Records of children (<18 years of age) admitted between 1 January 2015 and 31 December 2020 with a presumed infectious diagnosis were included. Patients who were empirically started on anti-MRSA antibiotics (i.e., vancomycin, clindamycin, linezolid, or trimethoprim/sulfamethoxazole) had a PCR-based MNS (MAX StaphSR Assay, BD, Franklin Lakes, NJ, USA), and a clinical culture performed were retrospectively reviewed. Excluded were patients who had obtained MNS as part of a surveillance screen during treatment in the neonatal intensive care unit or prior to admission for surgery, MNS obtained >5 days after admission, and MNS performed >48 h after initiation of anti-MRSA antibiotics. Furthermore, we excluded patients who were started on antibiotics as part of a 48 h sepsis rule out. The collected data included age, gender, timing (in relation to admission date and to start date of anti-MRSA antibiotics) and the result of MNS test, clinical diagnosis, the result of microbiologic cultures, anti-MRSA antibiotics used, start and discontinuation date of anti-MRSA antibiotics, and documented reason for discontinuation of anti-MRSA antimicrobials. 

Descriptive statistics were used to describe the cohort using medians and ranges for continuous variables and frequencies and percentages for categorical variables. The sensitivity, specificity, PPV, and NPV and their exact 95% confidence intervals (95% CI) were calculated in order to evaluate the use of the MNS in predicting MRSA in a clinical culture for the whole cohort. Duration of anti-MRSA antibiotic use among those who had stopped its empiric use before hospital discharge were compared by clinical reason for discontinuation using nonparametric Kruskal–Wallis rank sum test. A *p*-value of <0.05 was considered statistically significant. Statistical analysis was performed using SAS version 9.4 (SAS Institute, Inc., Cary, NC, USA).

## Figures and Tables

**Table 1 antibiotics-10-01434-t001:** Demographic characteristics, diagnosis categories, and anti-MRSA antimicrobials used.

Variables	All, *n* = 95
**Median age (range), years**	2 (0–17)
**Gender, *n* (%)**	
Male	55 (57.9)
Female	40 (42.1)
**Diagnosis, *n* (%)**	
Skin and soft tissue infections	38 (40)
Toxin-mediated syndromes	17 (17.9)
Osteoarticular infections (OAI)	14 (14.7)
Pneumonia/pleural effusion (PNA)	6 (6.3)
Oto/mastoiditis	6 (6.3)
Other diagnoses (each n < 5) ^1^	14 (14.7)
**Anti-MRSA antimicrobials** **empirically started, *n* (%)**	
Clindamycin	50 (52.6)
Vancomycin	38 (40.0)
Clindamycin + vancomycin	5 (5.3)
Linezolid	1 (1.1)
Trimethoprim-sulfamethoxazole	1 (1.1)

^1^ retropharyngeal abscess, *n* = 4; lymphadenitis, *n* = 3; epiglottitis, *n* = 2; parotitis, *n* = 2; tracheitis, *n* = 1; bacteremia, *n* = 1; sepsis from bowel perforation, *n* = 1.

**Table 2 antibiotics-10-01434-t002:** Clinical culture results by diagnosis category.

Culture Result ^1^, n (%)	All, *n* = 95	SSTI ^2^, *n* = 38	Toxin-Mediated Syndrome, *n*= 17	OAI ^3^, *n* = 14	PNA ^4^, *n* = 6	O/M ^5^, *n* = 6	Other ^6^, *n* = 14
Any growth	53, (55.8)	21, (55.3)	12, (70.6)	5, (35.7)	2, (33.3)	4, (66.7)	9, (64.3)
Methicillin-sensitive *Staphylococcus aureus* (MSSA)	23, (24.2)	10, (26.3)	6, (35.3)	4, (28.6)	0	0	3, (21.4)
*Streptococcus* spp.	14, (14.7)	9, (23.7)	2, (11.7)	0	0	1, (16.7)	2, (14.3)
Methicillin-resistant *Staphylococcus aureus* (MRSA)	7 (7.4)	1, (2.6)	2, (11.7)	0	0	0	4, (28.6)
Gram-negative rods ^7^	10 (10.5)	5, (13.2)	0	1, (7.1)	1, (16.7)	0	3, (21.4)
Coagulase-negative *Staphylococcus* spp.	6, (6.3)	2, (5.3)	2, (11.8)	0	0	2, (33.3)	0
*S. pneumoniae*	2, (2.1)	0	0	0	1, (16.7)	1, (16.7)	0
*E. faecalis*	2, (2.1)	0	1, (5.9)	0	0	0	1, (7.1)

^1^ Data shown indicates the number of individuals with positive clinical cultures; individuals with >1 bacteria recovered from culture (*n* = 13) were noted, mostly in cases with SSTI (*n* = 6). MSSA and MRSA positive cultures with co-bacteria were found (*n* = 5 and *n* = 4, respectively); ^2^ skin and soft tissue infections; ^3^ osteoarticular infections; ^4^ pneumonia; ^5^ oto/mastoiditis; ^6^ other includes retropharyngeal abscess, *n* = 4; lymphadenitis, *n* = 3; epiglottitis, *n* = 2; parotitis, *n* = 2; tracheitis, *n* = 1; bacteremia, *n* = 1; sepsis from bowel perforation, *n* = 1; ^7^
*H. influenzae* (*n* = 2), *Enterobacter* (*n* = 1), *P. aeruginosa* (*n* = 2), *E. coli* (*n* = 2), *K. pneumoniae* (*n* = 1), *K. oxytoca* (*n* = 1), *Stenotrophomonas* spp. (*n* = 1), *M. morganii* (*n* = 1), *Acromobacter* (*n* = 1).

**Table 3 antibiotics-10-01434-t003:** Nasal MRSA^1^ screening results vs. clinical culture results.

Nasal Screening	Culture Results		
	MRSA ^1^ +	MRSA −	Total
MRSA +	3	4	7
MRSA −	4	84	88
Total	7	88	95

^1^ methicillin-resistant *Staphylococcus aureus*.

**Table 4 antibiotics-10-01434-t004:** Test characteristics of nasal screening to predict clinical infection with MRSA ^1^.

Variable	n/N	%	95% CI
Sensitivity	3/7	42.9	9.9–81.6
Specificity	84/88	95.5	88.8–98.8
Positive predictive value (PPV)	3/7	42.9	9.9–81.6
Negative predictive value (NPV)	84/88	95.5	88.8–98.8

^1^ methicillin-resistant *Staphylococcus aureus.*

## Data Availability

Not applicable.

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
