# Peer review of "Utility of Methicillin-Resistant Staphylococcus aureus Nares Screening in Hospitalized Children with Acute Infectious Disease Syndromes"

_antibiotics, 2021, doi:10.3390/antibiotics10121434_

Round 1

Reviewer 1 Report

The present communication reported the utility of Methicilin resistant Staphylococcus aureus nares screening in hospitalized children with acute infectious disease syndromes. Of 95 children screened, seven patients (7.4%) were found positive for both Nasal MRSA colonization and growth of MRSA in clinical cultures. The specificity and the negative predictive value (NPV) of the MRSA Nare Screening to predict a clinical MRSA infection were both 95.5%. The manuscript is well conceived and developed. Minor revisions needed. Following the comments to the paper:

Line 67: insert the number of female analyzed in the study both in the text and in Table 1.

Line 78-79: The prevalence of MSSA, based on 18 positive sample, is 19% instead of 34%, indicated in the text. Moreover, the prevalence of MRSA is 7.4% (7/95) instead of 13.2% indicated in the text.

Table 2: Please indicate the percentage with the relative ratio, especially in the line of “(+) growth”. For example:  21/38 (55.3%).

In the polimicrobial culture results there are three sample also positive to MSSA which are not considered in MSSA group. So the prevalence of MSSA is 22,1% (21/95).

Table 4: Please verify the value of NPV, and indicate below the table the significance of “n” and “N” in the formula “n/N”.

Author Response

Line 67: insert the number of female analyzed in the study both in the text and in Table 1.

Thank you for this suggestion. We have delineated female patients as recommended.

Line 78-79: The prevalence of MSSA, based on 18 positive sample, is 19% instead of 34%, indicated in the text. Moreover, the prevalence of MRSA is 7.4% (7/95) instead of 13.2% indicated in the text.

Thank you for this correction. We have adjusted the text to match the values included in Table 2.

Table 2: Please indicate the percentage with the relative ratio, especially in the line of “(+) growth”. For example:  21/38 (55.3%).

Thank you for this suggestion, however we find that adding the relative ratios will decrease the readability of this table.

In the polimicrobial culture results there are three sample also positive to MSSA which are not considered in MSSA group. So the prevalence of MSSA is 22,1% (21/95).

Thank you for this suggestion. We have distributed all recovered pathogens to individual rows to be sure to include all appropriate pathogens including MSSA. In re-reviewing the data, we included four additional isolates that were recovered in addition to MRSA.

Table 4: Please verify the value of NPV, and indicate below the table the significance of “n” and “N” in the formula “n/N”.

Thank you for pointing out this error. We have corrected the n/N for NPV included in the table.

Reviewer 2 Report

 On account of the manuscript ANTIBIOTICS-1424943, entitled “Utility of Methicillin-resistant Staphylococcus aureus nares screening in hospitalized children with acute infectious disease syndromes” by Ashley Sands et al., the authors investigated the utility of nares screening (MNS) for anti-MRSA antibiotic de-escalation in hospitalized children based on a PCR-based MNS, and a clinical culture performed. The topic is important to better understanding of the effectiveness of MNS for specific infectious diagnoses, and the authors got interesting results. The manuscript was well written and designed. After careful consideration, I made a decision that the manuscript is acceptable for publication in its present form.

Special remarks:

‧ The present manuscript retrospectively evaluated the utility of MNS in hospitalized children with presumed infectious conditions, with the goal of comparing MNS with clinical culture results.

S. aureus and MRSA-colonization rates were important to pediatric patients who receiving care in the neonatal intensive care environment, as well as pre-operatively. On the other hand, studies evaluating MNS in children admitted for presumed infectious disease diagnoses are still limited.

‧ The present Communication provide the insight for the effectiveness of MNS for specific infectious diagnoses.

‧ The interpretation of the evidence and arguments presented and conclusions are sufficient.

‧ The references cited relevant and up to date.

‧ The tables are useful, necessary, and good quality.

Author Response

Thank you for your thoughtful review and comments.

Reviewer 3 Report

The paper by Sands et al. attempts to show the utility of methicillin-resistant Staphylococcus aureus nares screening in children.  Overall, the paper is well written and has used the appropriate statistical tools.    However, the main issue is the sample size is too small to state a conclusion.  Only seven cultures ended up being MRSA.  The nasal screening MRSA+ row had only three that were MRSA+ and four that were MRSA-.  The MRSA- nasal screening only had four culture positive MRSA+.  A greater number of samples with MRSA in them is needed to make any definitive claim.

Minor issues to address:

  1. Line 78 should be methicillin-sensitive.
  2. Line 140 In this context,
  3. Lines 140-147  Break up this sentence. It was hard to follow.
  4. Line 147 The difference, however, did not reach statistical significance, which.
  5. Italicize the species names in the reference section.
  6. Two different styles were used for the references.

Author Response

The paper by Sands et al. attempts to show the utility of methicillin-resistant Staphylococcus aureus nares screening in children.  Overall, the paper is well written and has used the appropriate statistical tools.    However, the main issue is the sample size is too small to state a conclusion.  Only seven cultures ended up being MRSA.  The nasal screening MRSA+ row had only three that were MRSA+ and four that were MRSA-.  The MRSA- nasal screening only had four culture positive MRSA+.  A greater number of samples with MRSA in them is needed to make any definitive claim.

Appreciated. We pointed this limitation out in our discussion. For this reason, we are not making any definitive claims and instead ask for additional prospective studies to be conducted.

Minor issues to address:

  1. Line 78 should be methicillin-sensitive.
  2. Line 140 In this context,
  3. Lines 140-147  Break up this sentence. It was hard to follow.
  4. Line 147 The difference, however, did not reach statistical significance, which.
  5. Italicize the species names in the reference section.
  6. Two different styles were used for the references.

Thank you for these corrections. All revisions have been made as suggested.

Round 2

Reviewer 3 Report

As mentioned before, the sample size is too small.